# Differential Effects of Proactive and Reactive Work Connectivity Behavior After-Hours on Well-Being at Work: A Boundary Theory Perspective

**DOI:** 10.3390/bs15030320

**Published:** 2025-03-06

**Authors:** Lingling Li, Guanfeng Shi, Xiong Zheng

**Affiliations:** School of Economics and Management, Shihezi University, Shihezi 832000, China; lingling4888@163.com (L.L.); shigf0109@163.com (G.S.)

**Keywords:** proactive work connectivity behavior after-hours, reactive work connectivity behavior after-hours, job control, work-to-home conflict, integration preference, well-being at work

## Abstract

This study examines the differential impact of proactive and reactive work connectivity behaviors on job well-being, drawing from the perspective of boundary theory. The increasing popularity of work connectivity behavior after-hours (WCBA) has attracted widespread attention from scholars on the relationship between WCBA and employee well-being. One view suggests that the impact of WCBA is negative, while another view suggests it is positive. Obviously, the impact of (WCBA) on well-being at work is still contradictory. To clarify the complexity of the impact of WCBA on well-being at work, based on boundary theory, we divided WCBA into proactive WCBA (PC) and reactive WCBA (RC), and examined the double-edged sword effect of WCBA on well-being at work, as well as the mediating mechanisms of job control and work-to-home conflict, and the moderated effects of boundary segmentation preferences. This study uses an empirical sampling method to collect data from 125 employees for a period of five days for quantitative research. The results show that, first, PC has an inverted U-shaped effect on job control, and job control is negatively related to well-being at work. Thus, the mediating effect of job control is significant. Second, RC has a negative impact on job control, and there is also a negative relationship between job control and well-being at work. Therefore, the mediating effect of job control is significant. Third, PC and RC are positively correlated with work-to-home conflict, and work-to-home conflict has a significant positive impact on well-being at work. Therefore, the mediating effect of work-family conflict is significant. Fourth, the study also found that integration preference moderates the relationship between work-to-home conflict and PC on well-being at work; that is, the mediating effect of work-to-home conflict is stronger for employees with a low integration preference. This study enriches our understanding of WCBA, PC, and RC from the perspective of proactive and passive employee behaviors. The study also provides a new interpretation of the impact of WCBA on well-being at work and offers valuable insights into sustainable development in digital social transformation and the application of boundary theory and the theory of empowerment–subjugation in achieving the United Nations’ Sustainable Development Goals. In addition, the study deepens our understanding of the heterogeneous regulatory role played by work–family integration preferences in influencing well-being at work under different types of WCBA.

## 1. Introduction

With the rapid development of information and communication technology (ICT), the traditional office mode has altered profoundly. Global enterprises have initiated a wave of digital transformation that hastens the evolution of office modes ([28]). Increasingly, enterprises are integrating digital communication devices and telecommuting software into daily organizational operations, blurring the boundary between work and personal time for employees ([41]). Employees are increasingly using portable communication tools (such as cell phones and tablets) to engage in work activities or connect with colleagues outside of regular work hours. [29] ([29]) termed this phenomenon non-working-time work-connected behavior or work connectivity behavior after-hours (hereinafter referred to as WCBA) ([12]). WCBA has blurred the boundaries between work and personal life ([38]). The encroachment of work tasks into non-work domains is increasingly prevalent. Consequently, employees often exhibit mixed feelings and attitudes towards WCBA ([41]). On the one hand, WCBA can help employees complete their work more flexibly and efficiently ([12]), improving work efficiency and satisfaction ([25]; [33]). On the other hand, excessive WCBA may hinder psychological disengagement ([32]), impede recovery from work ([10]), and even lead to work–family conflict ([35]), thereby adversely affecting job performance and attitudes ([39]; [20]). In particular, significant debate remains regarding the impact of WCBA on well-being at work ([12]; [39]). Some argue that WCBA has a positive impact on well-being at work., while others think it has a negative impact ([25]; [39]). Therefore, the main aim of this study was to clarify the effect of WCBA on well-being at work.

Well-being at work refers to employees’ psychological experiences, satisfaction levels, and emotional states related to their work ([40]). Well-being at work is crucial for the sustainable development of society and is a shared goal for both organizations and employees. High levels of well-being generate performance improvements. Therefore, how to enhance employees’ well-being at work is a critical management practice issue. Some studies have shown that WCBA complicates psychological detachment ([29]) and extends working hours and workloads ([20]; [11]), impacting recovery experiences ([32]; [3]), and thereby reducing well-being at work ([25]). Other studies have pointed out that WCBA may enhance employees’ control over their work ([18]), increase job flexibility and autonomy ([6]), and foster more positive perceptions ([37]), thereby enhancing well-being at work ([12]). Consequently, conflicting findings exist regarding the impact of WCBA on well-being at work, and its intricate mechanisms remain incompletely understood ([39]; [35]). Therefore, clarifying the influence mechanism of WCBA on well-being at work could help us determine mitigation methods for the adverse effects of WCBA.

This study has four research aims and examines the mediation path and moderating mechanism between WCBA and well-being at work in both the work and family domains.

First of all, according to the boundary theory ([8]), this study examines the impact of WCBA on well-being at work, specifically dividing WCBA into PC and RC for the behavioral aspect. Based on the research of [12] ([12]), this study categorizes WCBA into two types: proactive non-working-time work-connected behavior (PC, e.g., an employee came up with a good work idea while relaxing at home, so they gave up resting and went online to connect with work) and reactive non-working-time work-connected behavior (RC, e.g., an employee received an instant message from their leader during dinner, so they interrupted family activities and went to work according to their instructions). However, most existing studies have not distinguished between the types of WCBA ([25]; [37]). Thus, the study compares and analyzes the heterogeneous impacts of these two types of WCBA on well-being at work, aiming to provide explanations for the contradictions found in previous research.

Secondly, this study tests the inverted complex effect of PC on job control and well-being at work, offering a unique perspective on understanding the relationship between PC and well-being at work.

Thirdly, the study comprehensively considers the two mediating mechanisms of the work and family domains when analyzing the effects of WCBA. The existing research often chose to study the impact of WCBA on well-being at work through focusing a single path in the family or work domain ([14]; [4]). Therefore, this study avoids the limitation of considering only one domain to gain a deeper understanding of the complex impact of WCBA on well-being at work.

Finally, this study examines the boundary conditions of work-to-home conflict. Some individuals tend to blur boundaries, and even prefer boundary integration, while others favor clear and distinct boundaries and disapprove of boundary-crossing ([18]; [19]). Therefore, family boundary integration preferences were selected for this study as a moderator to examine how WCBA affects work-to-home conflict. This contributes to a deeper understanding of individual differences in coping with RC.

## 2. Literature Review and Hypotheses

There have been several related achievements regarding the subdivision of WCBA and related explanatory theories, such as the classification of WCBA into PC and RC, which can be interpreted using the boundary theory to understand the influence of WCBA ([12]). Reviewing these studies can help us understand the reasoning behind the hypotheses of this study.

### 2.1. PC and RC

PC refers to employees actively using work-related communication tools to engage in work during non-working hours without receiving work instructions from the workplace (including supervisors, colleagues, and clients). RC refers to employees using work-related communication tools to engage in work during non-working hours in response to work instructions from the workplace (including supervisors, colleagues, and clients). From the perspective of behavioral occurrences, PC involves employees spontaneously altering the permeability or flexibility of their psychological boundaries first, before actively adjusting the permeability or flexibility of their physical and temporal boundaries. RC, on the other hand, entails other members of the work domain utilizing mobile technologies (such as smartphones and instant messaging apps) to breach the physical (contacting employees at home rather than at the office) and temporal boundaries (contacting employees during non-working hours) to enter an employee’s home domain, thereby altering the permeability or flexibility of that employee’s psychological boundaries ([8]; [26]). The former involves autonomously initiated cross-boundary actions, while the latter involves passively responding to a cross-boundary work demand ([8]). Thus, the underlying drivers of these behaviors differ fundamentally ([3]), leading to varied impacts on employees’ subsequent attitudes and emotions regarding work ([37]; [4]). Consequently, categorizing WCBA types is crucial for understanding why these behaviors can elicit both positive and negative effects on attitudes and emotions toward work.

### 2.2. WCBA and Well-Being at Work from the Perspective of Boundary Theory

Clark believes that boundary theory mainly emphasizes how changes in the boundary between work and family, as well as the relationships between employees who act as boundary-crossers and the people they come into contact with in different fields, affect work and family ([8]). Specifically, the boundary between home and work can be distinguished through four types of boundaries. Firstly, physical boundaries refer to the spatial range of actions in different domains. Secondly, the time boundary, in simple terms, refers to the start and end times of work. Thirdly, social boundaries mainly refer to the distribution of social relationships within a certain field. Fourthly, psychological boundaries refer to the individual’s identity within their role, as well as the thinking patterns and emotional expressions formed on this basis. Moreover, these boundaries usually have three characteristics: permeability, flexibility, and mixing ([8]).

From the perspective of boundary theory ([8]), WCBA is a typical cross-border behavior that blurs the physical, temporal, social, and psychological boundaries between work and family for individuals ([12]). For example, when engaging in work behavior after work, the participation in both a superior–subordinate relationship and a marital relationship reflects the permeability and flexibility of boundaries, which vary among individuals. Some individuals tend to integrate work and family, while others tend to separate the two to avoid interference and conflict ([19]). Boundary-crossing activities can allow for emotions, behaviors, and values in both the work and family domains to permeate and influence each other, which can be positive or negative ([41]). The rewards and resources obtained by individuals through work can help improve and assist in family life, which is beneficial for enhancing individual happiness and satisfaction in family life ([38]). The care and love that individuals receive from loved ones in their family can help them maintain good emotions, a positive state of mind, and vitality in their work ([38]). On the contrary, when work pressure troubles individuals, they may not be able to meet the needs of their family roles, leading to negative emotions towards work ([12]). Therefore, boundary theory is a fundamental theoretical perspective that is very suitable for studying the changes in individual psychological activities caused by cross-border behavior. Specifically regarding WCBA and well-being at work, we believe that the complex relationship between the two can be interpreted through boundary theory.

According to the perspective of boundary theory that boundaries have permeability and flexibility ([8]), WCBA will have an impact on both work and family, and the impact on family will, in turn, affect attitudes and emotions related to work ([12]). Therefore, when exploring the impact path of WCBA on well-being at work, it is necessary to consider both work and family aspects simultaneously. From the perspective of the job path, job control reflects the degree of self-control that employees have when engaging in work behavior across boundaries in the family domain ([18]). WCBA can help employees better control their work ([10]), and job control can improve employee satisfaction, reduce emotional exhaustion, and lead to positive emotions, thereby enhancing well-being at work ([12]; [38]). From the perspective of the family domain path, WCBA is an invasion of elements from the work domain into the family domain ([35]), which may disrupt the work–family balance and lead to conflict between work and families ([11]). Work conflicts and family conflicts can lead to negative attitudes and emotions towards work among employees, reducing well-being at work ([38]). Therefore, this study chose job control and work–family conflict as mediating pathways for the impact of WCBA on well-being at work in the work and family domains, respectively. Furthermore, based on the perspective of boundary theory, which suggests that the flexibility and permeability of boundaries vary among individuals ([35]; [8]), this study considers individuals’ boundary integration preferences as moderating variables for the mediating pathway of work conflict in families.

### 2.3. Work Domain Path: PC/RC, Job Control, and Well-Being at Work

Job control refers to the degree to which employees feel that their work can be controlled by themselves; the characteristics of a job can provide employees with freedom of choice. This control includes the ability to control time-planning, schedule arrangement, specific processes, material allocation, and other aspects ([18]). Job control plays an extremely key role in employees’ well-being at work ([12]). Through the use of work communication tools, employees can control the progress of their work and offer feedback anytime and anywhere, which is conducive to improving job control ([41]; [33]). However, constant work connectivity forces employees to always be online ([20]). In addition, employees passively respond to continuous calls from remote working environments ([6]), and they cannot fully control the frequency and time of their passive response ([39]), which greatly weakens employees’ autonomy over their schedule, working methods, and decision-making ([41]); that is, their job control is reduced. This contradiction is known as the paradox of WCBA (e.g., the fact that employees derive greater control from the use of technology at work, but at the same time, this control is eroded when employees are expected to be constantly connected) ([30]; [16]).

[12] ([12]) argue that the key to the paradox of WCBA lies in distinguishing between PC and RC. The modes in which PC and RC alter boundaries differ, which determines the types of interruptions they cause and, consequently, leads to differing impacts on job control ([34]). PC induces only internal interruptions within individuals, which cause employees to feel in control of their own volition, thereby providing a sense of mastery over their time and space ([12]). In contrast, RC constitutes an external interruption that often demands an immediate response, characterized by its unpredictability and enforceability and disrupting employees’ original schedule ([12]). Hence, PC enhances employees’ job control, whereas RC diminishes it.

However, when the PC exceeds a certain limit, employees also start to feel less in control of their job. If an employee is immersed in his or her work almost all day, he or she has actually become a “slave” to their work ([30]), being dominated and controlled by the work, and their flexibility and control of their work are greatly damaged ([23]). Constant online work can result in employees not being able to make autonomous decisions about work schedules, work methods, decision-making, etc. Employee autonomy at work is diminished ([1]). Therefore, excessive PC will weaken job control. Thus, PC has an inverted U-shaped effect on job control ([12]). At low levels of PC, PC enhances employee job control. At high levels of PC, PC will instead weaken employees’ job control.

Job control reflects employees’ autonomy in managing boundaries ([18]). This autonomy can provide employees with more choices between the work and family domains and provide positive emotional and psychological resources to employees ([8]). Positive emotions can directly improve employees’ well-being at work ([40]) and positive psychological resources can also help employees obtain other resources, reduce the emotional exhaustion caused by work requirements consuming work resources ([41]), and thus increase employees’ well-being at work. A number of existing studies have confirmed that job control can improve well-being at work ([15]). In sum, PC and RC can affect well-being at work by affecting job control. Based on the above discussion, this study proposes the following hypotheses:

**H1a.** 
*Job control mediates the inverted U-shaped relationship between PC and well-being at work.*


**H1b.** 
*Job control mediates the negative impact of RC on well-being at work.*


### 2.4. Family Domain Paths: PC/RC, Work-to-Home Conflict, and Well-Being at Work

Work-to-home conflict refers to the extent to which individuals perceive that work interferes with their responsibilities and roles in the family and causes their time and energy in the family domain to shrink ([13]; [22]). Specifically, the work-to-home conflict is divided into three aspects: time conflict, pressure conflict, and behavior conflict ([22]). Time conflict refers to the time spent at work, which makes it difficult for individuals to participate in family activities and complete family duties. Pressure conflict means that work pressure spills over to the family field, and individuals are still psychologically focused on work at home, so that it is difficult to meet family needs. Behavioral conflict is when behavior that is beneficial for work contradicts the behavior that is expected in the family.

WCBA increases three aspects of work-to-home conflict: time, recovery and conversion. In terms of time, work connectivity is reflected in the fact that employees are still working after regular work hours, which occupies the time originally spent with family ([21]), thus resulting in a time conflict. In terms of recovery, WCBA hinders employees’ psychological detachment ([20]; [9]), resulting in a compression of employees’ normal rest time and the failure to achieve an effective recovery experience ([2]). Over-consumed resources that cannot be replenished in time will lead to insufficient resources ([7]), resulting in stress ([36]) and stress conflicts. In terms of switching, the intermittent nature of WCBA requires individuals to constantly switch between the two domains of home and work ([8]). After work, individuals must always pay attention to work information, and the priority of the work role is usually higher than that of the family ([39]). This can cause individuals to be absent-minded when engaging in family role behaviors ([41]; [35]). This behavior, although beneficial for work, fails to meet the expectations of family role behavior ([38]) and thus may lead to behavioral conflicts.

More precisely, we argue that RC induces employees to have more work-to-home conflict relative to PC. From the perspective of employees’ psychological preparation, employees engaged in PC spontaneously increase their boundary-crossing to the work field in advance, and are psychologically prepared to work from home after work. Employees engaged in RC have difficulty anticipating when job demands will occur and are often asked to respond while unprepared ([12]). This difference has three effects: Firstly, employees engaged in PC can arrange the specific time of their WCBA in advance, which can buffer the adverse impact of WCBA on the family to a certain extent. However, employess engaged in RC lack time autonomy ([23]). Employees need to respond passively to the job’s demands. This leads to more time conflicts ([25]). Secondly, compared with employees engaged in PC, employees engaged in RC are usually not psychologically prepared to face the RC and face a greater threat of resource loss, so they feel greater work pressure ([12]; [36]). This can lead to more stressful conflicts. Thirdly, employees engaged in PC generally avoid PC that will interfere with important family activities ([36]). However, employees engaged in RC cannot predict when RC will occur, so RC will cause additional behavioral conflicts.

The negative impact of work-to-home conflict on well-being at work has been confirmed by some studies ([41]; [31]). From the perspective of time conflicts, if individuals often use the time that should be used in the family domain to work, the support provided by resources from the family domain will be reduced. This will affect the completion of subsequent work and lead to job burnout ([39]), which, in turn, reduces well-being at work. From the perspective of stress conflicts, when work stress spills over to the family domain, employees cannot recover the mental energy lost at work at home. This will lead to increased stress ([25]), which will lead to emotional exhaustion in subsequent work ([12]) and reduce happiness with the job. From the perspective of behavioral conflicts, if employees often engage in behaviors beneficial to work after work, they cannot meet the behaviors expected by family members. This will affect employees’ performance of their family roles and may cause dissatisfaction among family members ([38]), which will be transmitted to employees, who will further transfer such negative emotions to their attitudes toward work ([39]). Therefore, this reduces job happiness.

**H2a.** 
*PC and RC are negatively related to well-being at work through increased work-to-home conflict.*


**H2b.** 
*Compared to PC, RC leads to a greater amount of work-to-home conflict, ultimately resulting in a more pronounced reduction in well-being at work.*


Boundary integration preference refers to the degree of individual preference regarding the connection between affairs in the work domain and those in the home domain, and their willingness to minimize the boundary between the two ([19]). Integrators (i.e., employees who prefer to integrate work into their home domain) like to bring unfinished work to the family after coming home, and tend to reduce the boundary between work and family so that work affairs can flow to the family field ([19]). Integrators prefer their work and family boundaries to be highly flexible. Employees with a low integration preference prefer strict and impermeable boundaries between the two domains ([25]). WCBA is the penetration of work elements into the home domain, which uses boundary integration preferences to allow employees to continue completing work during their off-hours ([24]). This satisfies employees who prefer to work continuously at home ([24]). Therefore, for those with a strong boundary integration preference, the impact of WCBA on work-to-home conflict is alleviated ([37]). Employees with a low level of boundary integration preference do not want work matters to disturb their families ([39]), and when it work time encroaches on family leisure time, such employees are more likely to feel their work–family balance is disrupted and experience higher levels of work-to-home conflict ([11]), strengthening the effect of WCBA on work-to-home conflict. Therefore, individual boundary integration preferences are able to mitigate WCBA-induced work-to-home conflict.

More precisely, we argue that, relative to RC, the relationships in PC-induced work-to-home conflicts are affected to a higher degree by individual boundary integration preferences. According to the boundary theory ([8]), in addition to individual boundary integration preferences, the work-to-home conflict also depends on employees’ autonomy in boundary management ([25]). Employees with more autonomy in boundary management experience less work-to-home conflict ([12]). Employees take the initiative to manage their boundaries and are more able to make independent choices regarding boundary management. Therefore, compared with employees engaged in RC, employees engaged in PC have greater autonomy in boundary management. Work-to-home conflict is more easily mitigated by personal boundary integration preferences. Based on this argument, this study proposes the following hypothesis:

**H3a.** 
*The negative impact of PC (RC) on well-being at work through work-to-home conflict is mitigated by integrators.*


**H3b.** 
*Relative to RC → work-to-home conflict → well-being at work, the PC relationship → work-to-home conflict → well-being at work is more strongly mitigated by individual integration preferences.*


According to the hypotheses, this study proposes the theory model shown in Figure 1.

## 3. Materials and Methods

### 3.1. Sample and Procedures

The subjects of such studies were largely knowledge-based employees engaged in digital work, such as employees in the financial and Internet industries. Therefore, this study selected MBA students from the finance, software services, and online education industries as participants, primarily from the business schools of five universities in China. The participants were in full-time work during normal working hours, working in a regular office. The study of MBA courses was conducted during holidays and weekends.

Most of the variables involved in this research exhibited significant fluctuations at the daily level. For instance, there were notable differences in the time and frequency employees spend participating in WCBA each day, leading to corresponding variations in related mediating and outcome variables. To better capture this variability, the study draws upon the latest research methods and designs, particularly the increasingly popular longitudinal survey method known as the Experience Sampling Method (ESM) ([12]). Essentially, ESM is a diary-based survey research approach that allows for the collection of questionnaire data from the same group of subjects over a continuous period, thereby obtaining relatively accurate and timely feedback from participants. Previous studies show that assessments conducted over a consecutive week can adequately reflect subjects’ fluctuations in the relevant variables ([12]). Furthermore, to avoid imposing additional burdens on employees during non-workdays, we conducted the survey only on weekdays (five consecutive days). The specific procedures we undertook for our investigation are outlined as follows.

First, the research team members contacted potential participants via email and mobile phone to schedule appointments for voluntary participation in the study. All participants were required to answer whether they experienced WCBA prior to joining the formal investigation; those who responded “no” were not invited to participate. Second, to facilitate simultaneous participation from subjects in different regions, the research team created a temporary anonymous WeChat group to invite subjects to take part in the survey. At the commencement of the formal study, we collected and coded the participants’ basic information through a questionnaire, allowing for the seamless matching of subsequent questionnaires over the following five days. The pre-prepared electronic questionnaire links were then disseminated to the group at different time intervals, with participants instructed to complete the questionnaires between 21:00 and 22:00. Reminders were sent at 21:30 and 21:45 to those who had not yet completed the survey. Finally, with the help of individual information and coding data, we conducted a matching process at the individual level for all completed questionnaires, resulting in a daily survey record for each participant. Notably, the Integration Preference is a relatively stable individual characteristic with minimal daily fluctuations; therefore, the Integration Preference Scale was only required to be completed on the first day, and the data obtained on that day were used for the remaining four days.

Initially, a total of 150 employees were invited to voluntarily participate in the survey. However, subjects that did not complete the basic information-matching, those who withdrew early, and subjects with evident response patterns were excluded. Ultimately, 125 participants completed a total of 625 valid daily survey questionnaires, resulting in an overall valid response rate of 83.3%. This study is a longitudinal study that primarily involved a five-day survey of 125 participants. Among the 125 participants, 75 were females, ranging in age from 21 to 56 years old (M = 29.78, SD = 6.71), with the time spent in their current organization ranging from 1 year to 12 years (M = 6.72, SD = 7.27), and working hours per week ranging from 40 to 60 h (M = 45.98, SD = 7.33).

### 3.2. Measurements

The variables involved in this study were measured using scales widely used in previous studies, and some issues were slightly revised based on the actual research situation. PC and RC were scored using a 6-point Likert scale, while the remaining variables were scored using a 7-point Likert scale.

The PC and RC Scale were adapted from the WCBA scale developed by [11] ([11]) and [12] ([12]), and this study followed this approach. In order to distinguish between PC and RC, this study added the time and frequency with which respondents were asked whether they actively initiated or were required to engage in work activities using smartphones and computers during off-hours. The WCBA, PC, and RC Scales each have four items. One of the items on the WCBA scale is “How many times did you use your computer for work outside of work hours today?”. After adapting the item, the corresponding examples for the PC and RC scales were “How many times have you actively used a computer for work outside of work hours today” and “How many times have you been asked to use a computer for work outside of work hours today”. The value of Cronbach’s alpha for the PC was 0.83; The value of Cronbach’s alpha for the RC was 0.84.

Job control was measured using a four-item scale developed by [18] ([18]). A sample item is “I can get through difficult times at work because I have experienced before difficulties”. The value of Cronbach’s alpha for job control was 0.83.

Work-to-home conflict was measured using a six-item scale developed by [5] ([5]). A sample item is “My work keeps me from my family activities more than I would like”. The value of Cronbach’s alpha for organizational support was 0.87.

Integration preference was measured using a four-item scale developed by [19] ([19]) and [18] ([18]). A sample item is “I am immersed in my work”. The Cronbach’s alpha was 0.84.

Well-being at work was measured using a six-item scale developed by [40] ([40]). A sample item is “I find real enjoyment in my work”. The Cronbach’s alpha for this scale was 0.88. 

### 3.3. Confirmatory Factor Analysis and Common Method Variance

This study selected Mplus 8.3 for multi-level confirmatory factor analysis (CFA). The results after comparing multi-level CFA result models indicated that the seven-factor model, consisting of PC, RC, job control, work-to-family conflict, integration preference, and well-being at work (χ^2^/df = 2.38, CFI = 0.91, TLI = 0.90, SRMR = 0.06, RMSEA = 0.04), demonstrated a superior fit to the data compared to the other models. Therefore, the results indicated that the discriminant validity of the seven variables is good.

The results of Harman’s single-factor test showed that the first factor only explained 39.63% of the total variation, which is lower than 50% ([27]). Therefore, in this study, the common method variance problem was not significant.

Given that the research variables involve cross-level issues, in order to ensure the accuracy of the model, the variance among the variables was differentiated within and between individuals before the hypothesis was validated. The results, presented in Table 1, show that the ICC (1) of each variable ranged from 0.71 to 0.83, indicating that this study is suitable for multi-level analysis. The proportion of variance within an individual ranged from 17% to 29%, indicating fluctuations in key variables between days.

## 4. Results

### 4.1. Descriptive Statistics and Correlations

Table 1 presents the descriptive statistics and correlations of the variables. PC squared was negatively related to job control (r = −0.24, *p* < 0.001), indicating that the impact of PC on job control has a curve effect, and RC was negatively related to job control (r = −0.23, *p* < 0.001), with a positive correlation between job control and well-being at work (r = 0.28, *p* < 0.001). Additionally, PC was positively related to work-to-home conflict (r = 0.22, *p* < 0.001) and RC was positively related to work-to-home conflict (r = 0.32, *p* < 0.001), while work-to-home conflict was positively related to well-being at work (r = −0.61, *p* < 0.001). This result is consistent with the theoretical expectations and provides preliminary support for the theoretical research.

### 4.2. Hypothesis Testing

When all control variables were included in the model and controlled for, this study tested the research hypothesis through multi-level path analysis. As shown in Figure 2, the square term of PC had a negative association with job control (β = −0.09, *p* < 0.01), indicating that PC had an inverted U-shaped effect on job control. Figure 3 shows how the sense of job control changes with the PC. Job control was positively associated with well-being at work (β = 0.23, *p* < 0.01). The mediating effect value of PC → job control → well-being at work was 0.02; furthermore, the Monte Carlo simulation showed that the 95% confidence interval did not include 0 ([0.002,0.05]), indicating that job control mediates the inverted U-shaped relationship between PC and well-being at work and supporting Hypothesis 1a.

As shown in Figure 2, RC was negatively related to job control (β = −0.15, *p* < 0.05), while job control was positively related to well-being at work (β = 0.25, *p* < 0.01); moreover, the mediating effect value of PC → job control → well-being at work was 0.04 and the 95% confidence interval did not include 0 ([0.003, 0.09]), indicating that job control mediated the positive correlation between RC and well-being at work. Thus, Hypothesis 1b was supported. The path coefficient of the RC square term affecting work control was not significant (β = −0.01, *p* = 0.135), indicating that RC and job control do not constitute a curve relationship, but only a linear relationship.

As shown in Figure 2, PC and RC were positively associated with work-to-home conflict (β = 0.19, *p* < 0.01; β = 0.27, *p* < 0.001) and work-to-home conflicts were positively associated with well-being at work (β = −0.28, *p* < 0.001; β = −0.34, *p* < 0.001); moreover, the mediating effect value of work–family conflicts between PC and work-to-home conflicts was −0.06, and the 95% confidence interval did not include 0 ([−0.08,−0.001]), while the mediating effect value of work–family conflicts between RC and work-to-home conflicts was −0.09, and the 95% confidence interval did not include 0 ([−0.11,0.001]). The difference in path coefficients between PC → work-to-home conflict and RC → work-to-home conflict was also significant (Δβ = −0.17, *p* < 0.05). Thus, Hypothesis 2a and 2b were supported.

As shown in Figure 2, the coefficient of the interaction term between PC and integration preference was significant (β = −0.07, *p* < 0.05), indicating that integration preference alleviated the effect of PC on work-to-home conflict. The moderated mediating effect value of PC → work-to-home conflict → well-being at work is −0.03, and the 95% confidence interval does not include 0 ([−0.04,−0.002]), indicating that integration preference moderated the mediating role of work-to-home conflict between PC and well-being at work. However, the coefficient of the interaction term between RC and integration preference was not significant (β = −0.04, *p* > 0.05), indicating that integration preference did not moderate the mediating role of work-to-home conflict between PC and well-being at work. Thus, Hypothesis 3a was partially supported, while Hypothesis 3b was supported. We speculate that the reason that integration preference cannot buffer the positive effect of RC on work-to-home conflict may be due to the unpredictability of RC, which makes it difficult for employees to plan their work-from-home time in advance, resulting in employees still perceiving higher levels of work-to-home conflict during periods of RC.

## 5. Discussion

The results of the multilevel path analysis indicate that PC can enhance well-being at work within a certain range by increasing job control. However, when the increase in PC exceeds this range, it may diminish job control and subsequently lower well-being at work. The findings suggest that even WCBA do not consistently improve job control, and therefore cannot perpetually enhance employees’ well-being. Conversely, when PC surpasses a specific threshold, WCBA may become burdensome, resulting in diminished job control and a decline in well-being. Additionally, the research indicates that RC can adversely affect well-being at work by reducing job control. Furthermore, both PC and RC are shown to decrease well-being at work by fostering work-to-home conflict. Consequently, this study elucidates the reasons behind the inconsistencies in existing research regarding the relationship between WCBA and well-being at work00. Such inconsistencies may stem from variations in WCBA types, differences in the underlying mechanisms, or the non-linear nature of the relationship between PC, mediator variables, and well-being at work. Thus, WCBA can be either a facilitative tool that enhances job control and yields benefits or a hindrance that induces work-to-home conflicts and has a detrimental effect. The root of this contradiction lies in the types of WCBA and the degree of engagement with each type. Moreover, the research findings reveal that integration preferences can mitigate the mediating effect of work-to-home conflict regarding PC and well-being at work. This outcome aligns with the existing literature; however, previous studies primarily examined this issue through the lens of resource conservation theory ([41]; [38]), while this study derived similar conclusions via boundary theory ([8]).

### 5.1. Theoretical Contributions

First, based on boundary theory, we subdivided WCBA into PC and RC, responding to scholars’ calls for a clearer delineation of WCBA and related constructs ([3]). Simultaneously, we further elaborated on the paradox of WCBA with specific empirical studies, elucidating the double-edged sword effect of WCBA on well-being at work. Although previous research explored the relationship between WCBA and well-being at work ([4]; [15]; [17]), the lack of prior differentiation among WCBA’s categories may have been a significant reason for the inconsistent findings in earlier studies. From an empowering perspective, PC can empower employees, thereby yielding positive outcomes. Conversely, from an enslaving perspective, excessive PC and RC are perceived as a burden, leading to adverse consequences ([30]; [16]). Therefore, this study offers a novel interpretation of the “empowerment–enslavement” paradox, enriching the understanding of WCBA, PC, and RC from the perspective of the “empowerment/enslavement” paradox of technology.

Secondly, this study examines well-being at work as the outcome variable in response to the call to consider well-being at work as a desired outcome rather than a mediating variable ([40]). Previous research often treated well-being at work as a mediating factor in organizational and employee performance, but there is a new trend in research to view well-being at work as the ultimate goal for both organizations and employees ([17]). This study follows this new trend by specifically linking WCBA to well-being at work through the use of boundary theory and the “empowerment/enslavement” paradox of technology. Thus, this study provides valuable insights regarding the application of boundary theory and the “empowerment/enslavement” paradox of technology in the sustainable development of the digital society, and finally to achieve ‘health and well-being at work’ (SDG-3).

Third, based on the boundary theory, this study introduces job control and work-to-home conflict as significant variables in the work-domain path and family-domain path, revealing the multi-domain paths through which WCBA affects well-being at work, and thus enriching the application scope of the boundary theory. The existing literature indicates that WCBA is a typical behavior that crosses the boundaries of work and family life ([38]; [8]), and therefore its impact on well-being at work needs to consider both the work domain and the family domain in order to better understand the complexity of WCBA’s effect on well-being at work ([12]; [17]). This study shows that, in the work domain, PC and RC have heterogeneous effects on well-being at work through job control; only a moderate range of PC leads to an increase in job control, while exceeding a certain range reduces job control, thereby affecting well-being at work, and RC only leads to a decrease in job control, consequently decreasing well-being at work. In the family domain, PC and RC both lead to work-to-home conflict and subsequently affect well-being at work. Therefore, the beneficial effects of WCBA on well-being at work only exist in the work-domain path, and PC needs to be limited to within a certain range. In the family-domain path, PC and RC only have negative effects on well-being at work. This study supplements the previous research conclusions on the double-edged sword effect of WCBA, providing a unique understanding of the relevant literature on the relationship between WCBA and work attitudes and emotions.

Fourth, the study enriches the moderating effect of work–family integration preference on the impact of WCBA. Some previous studies often considered variables such as family support, work achievement, work–family integration preference, and responsiveness from superiors as moderators in the impact of WCBA ([41]; [12]; [38]; [11]). However, based on boundary theory, this study found that the moderating effect of work–family integration preference is not consistent in the impact of PC and RC, with work–family integration preference only buffering the effect of work-to-home conflict in PC, not in RC. This conclusion deepens our understanding of the heterogeneous moderating role of work–family integration preference in the impact of different types of WCBA on well-being at work.

### 5.2. Practical Implications

From the results of this study, two important management implications can be drawn. First, the results show that the positive effect of WCBA on well-being at work is realized through an increase in PC, but excessive PC can lead to a decrease in job control, resulting in reduced well-being at work. Therefore, organizations need to provide tools for employees to control their work, such as portable office and communication devices. However, it is important to note that the detrimental effects of excessive engagement in WCBA should not be overlooked, and employees should be reminded to control the duration of their PC. In response to the significant negative effects of RC, including negative impacts in both the family and work domains, organizations should minimize RC as much as possible and provide support and necessary compensation for employees when they need to engage in RC in emergency situations in order to reduce the multiple negative impacts of RC. Additionally, organizations can minimize the unpredictability of WCBA by establishing organizational norms to maximize the negative effects of WCBA on employees.

Second, we found that integration preference can reduce the adverse effects of PC on work-to-home conflict and subsequently buffer the negative impact on job satisfaction, but this buffering effect does not exist in the context of RC. Based on these research findings, organizations need to tailor their approach to provide opportunities and pathways for employees with a high preference for work–family integration to engage in PC. For employees with a low preference for work–family integration, even if they want to engage in PC, the cost is often greater and leads to greater work-to-home conflict, meaning that PC is not worth it. Therefore, organizations should try to minimize WCBA for employees with a low preference for work–family integration.

### 5.3. Limitations and Future Research

First, the study exclusively focused on investigating the moderating role of integration preference, an individual-level factor. However, there are other social support factors, such as organizational support, leadership support, colleague support, and family support, which also play a role in influencing the impact of WCBA. Therefore, future research should explore how these external supportive factors affect the relationship between WCBA and well-being at work, providing a more comprehensive understanding of the complexity of this relationship.

Second, in the research design, all variables except for demographic variables were measured consistently from 21:00 to 22:00 daily. However, this approach did not allow for separate measurements of independent variables, mediator variables, and outcome variables at different time points throughout the day. As a result, the full potential of the empirical sampling method in reducing method bias was not fully realized, despite the relatively low severity of common method bias in the study. Future research could benefit from measuring various types of variables at multiple time points to effectively minimize common method bias.

Third, MBA students from finance, software services, and online education industries were selected as participants in this study, so the results of this study may be limited to specific industries and younger working groups. Further research is needed to determine whether the same conclusions could be obtained in other industries or age groups.

## 6. Conclusions

This study elucidates the complex mechanisms through which WCBA influences work-related well-being at work, based on a classification of the types of WCBA (PC and RC) across both the work and family domains. The findings indicate that, on one hand, within the work domain, PC exhibits an inverted U-shaped effect on job control and well-being at work, while RC diminishes well-being at work by reducing job control. On the other hand, in the family domain, both PC and RC impair well-being at work by increasing conflict, with the mediating role of work-to-home conflict between PC and well-being at work being mitigated by integrated preferences. These results suggest that the impact of WCBA on well-being at work is exceedingly intricate; in other words, its influence pathways are diverse. Furthermore, a preliminary classification of WCBA types is essential for accurately analyzing their interrelations to better understand the mixed effects WCBA can have on individuals. The findings provide evidence that when employees engage in WCBA practices, a thorough understanding of their complexities, along with appropriate countermeasures, can mitigate the potential negative effects and even enhance well-being at work to some extent. In summary, as various organizations undergo digital transformation, employees inevitably become involved in WCBA. Within this context, this study significantly contributes to deepening our understanding of WCBA’s complexities and provides valuable insights for the theoretical development and practical implications of WBCA, aiming to promote well-being at work.

## Figures and Tables

**Figure 1 behavsci-15-00320-f001:**
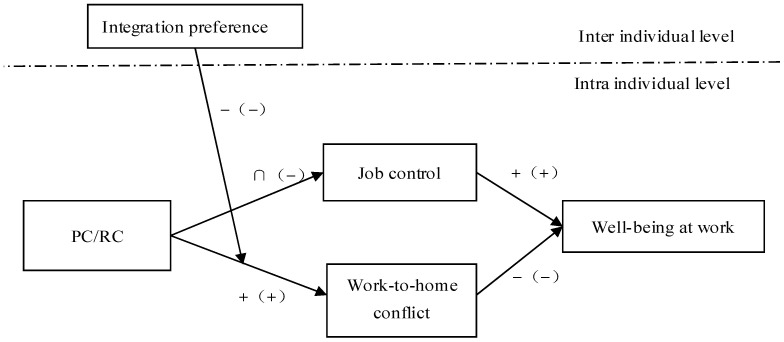
Theoretical model.

**Figure 2 behavsci-15-00320-f002:**
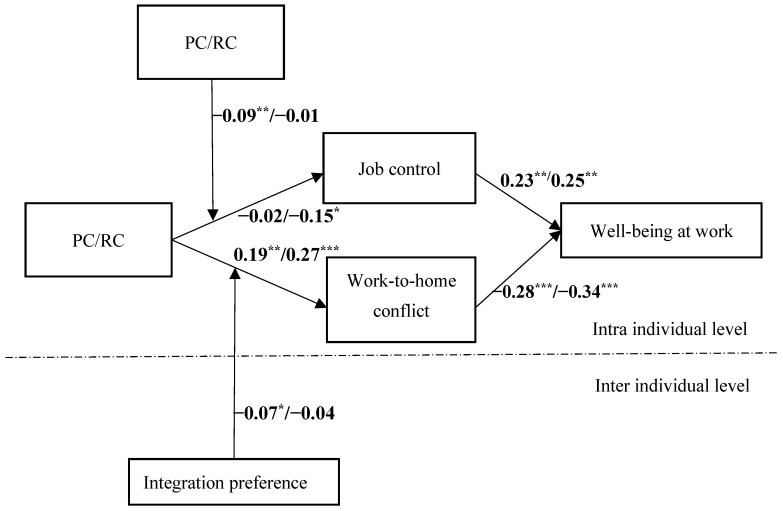
Multilevel path analysis results: PC/RC affects well-being at work. Note. * *p* < 0.05, ** *p* < 0.01, *** *p* < 0.001.

**Figure 3 behavsci-15-00320-f003:**
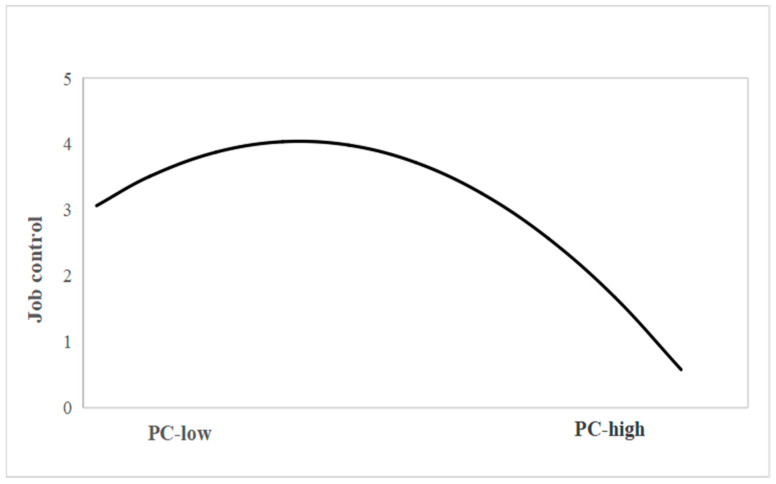
Curve effect of PC on job control.

**Table 1 behavsci-15-00320-t001:** The results of descriptive statistics, correlations and a multilevel test of variables.

Variables	M	SD	1	2	3	4	5	6	ICC(1) and Inter Individual Variance Percentage
Intra-individual level									
1. PC	3.19	1.48							0.72(28%)
2. RC	3.67	1.66	0.57 ***						0.71(29%)
3. JC	2.89	1.21	−0.15 *	−0.23 ***					0.78(21%)
4. WC	2.57	1.02	0.22 **	0.32 ***	−0.22 ***				0.74(26%)
5. WW	3.44	1.56	−0.35 **	−0.29 **	0.28 ***	−0.61 ***			0.83(17%)
6. PC × PC	2.26	2.37	0.36 ***	0.35 ***	−0.24 ***	0.14 *	0.15 **		
Intra-individual level									
1. GE	1.60	0.49							
2. AG	29.78	6.71	−0.19						
3. ED	2.46	0.85	−0.06	-0.10					
4. MA	2.61	2.51	−0.33 ***	0.17 **	0.44 ***				
5. WT	6.72	7.27	−0.13 *	0.24 ***	0.35 ***	−0.36 ***			
6. IP	2.83	0.64	−0.12 *	−0.11 *	0.07	−0.25 ***	−0.36 ***	0.01	

Note: JC, WC, WW, GE, AG, ED, MA, WT, and IP represent job control, work-to-home conflict, well-being at work, gender, age, education, marriage and childbirth, years of work experience, and integration preference, respectively. Note. * *p* < 0.05, ** *p* < 0.01, *** *p* < 0.001.

## Data Availability

The data presented in this study are available on request from the corresponding author.

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
