# Peer review of "Differential Effects of Proactive and Reactive Work Connectivity Behavior After-Hours on Well-Being at Work: A Boundary Theory Perspective"

_behavsci, 2025, doi:10.3390/bs15030320_

Round 1
Reviewer 1 Report
Comments and Suggestions for Authors
The article presents a thematic inserted in a theoretical framework that is pertinent to contemporary times and adjusted to organizational and social challenges, and related to the thematic scope of the: "Managing Organizational Behaviors for Sustainable Wellbeing at Work," achieving "empirically and theoretically discuss OB and wellbeing at work in the context of social sustainability and the UNSDGs."
“Empowerment or enslavement” mentioned in the title are not actually central topics in the research and so, despite making the title appetizing, the title should be more in line with the central content and variables of the article.
The article presents a pertinent and precise theoretical model. It presents a good articulation between the literature review and the hypotheses.
The variables were measured using scales widely used in previous studies. Optimizing their characterization and providing more information on the validity indicators of the instruments in the present study would be essential.
Summary tables for each instrument, highlighting their dimensional structure, would be important. The information presented in the results and relating to "Confirmatory Factor Analysis" should be integrated into the component relating to measures and instruments.
Hypothesis testing - with multilevel path analysis - is a vital research component.
In line 430, we suggest a structural change that separates the discussion components from the conclusion, fulfilling the specific objectives of each of these parts. The article lacks a practical discussion of the fundamental results, using counterpoint with bibliographical references mentioned or used in the first part.
Comments on the Quality of English Language"Minor editing of English language required"
Author Response
[Comments 1:[ “Empowerment or enslavement” mentioned in the title are not actually central topics in the research and so, despite making the title appetizing, the title should be more in line with the central content and variables of the article.]
Response 1: [Thank you for pointing this out. We agree with this comment. Therefore, we have deleted the word “Empowerment or enslavement” in the title]
Comments 2:[The variables were measured using scales widely used in previous studies. Optimizing their characterization and providing more information on the validity indicators of the instruments in the present study would be essential. ]
Response 2: [Thank you for pointing this out. We partially agree with this comment. The scales of variables involved in this study only partially optimized the characteristics of variables PC and RC. During the optimization process, we obtained permission from the original author, but we were required to keep the relevant details confidential. Because the project is an unfinished large-scale research project, all relevant personnel are required to keep it confidential. For this reason, we feel very embarrassed. After careful consideration, we are still unable to provide more details about the variables PC and RC. And the other scales used in this study were widely cited and published in previous research. This study did not optimize other variables, which are all single dimensional. As usual, the Cronbach’s alpha is provided. However, for the sake of simplicity in the article, we only provided this single indicator. The vast majority of research in this journal also follows this approach, so we referred to it. ]
[Comments 3:[Summary tables for each instrument, highlighting their dimensional structure, would be important.]
Response 3: [Thank you for pointing this out. We agree with this comment. However, this suggestion is not entirely appropriate in our research because all variables in our study are single dimensional and there are no multiple structural dimensions, so there is no need to emphasize their dimensional structure. For this reason, we did not make modifications to address this point.]
[Comments 4:[The information presented in the results and relating to "Confirmatory Factor Analysis" should be integrated into the component relating to measures and instruments.
Response 4: [Agree. Thank you for your recommendation. And we have made the necessary modifications based on your feedback and included the information related to "confirmatory factor analysis" provided in the results in the “Materials and Methods“section. ]
[Comments 5:[ In line 430, we suggest a structural change that separates the discussion components from the conclusion, fulfilling the specific objectives of each of these parts. The article lacks a practical discussion of the fundamental results, using counterpoint with bibliographical references mentioned or used in the first part. ]
Response 5: [ Agree. Thank you very much for your valuable feedback. Based on your suggestion, we have placed the conclusion section as a separate chapter after the research limitations and future studies, as the conclusion of the article. We reorganized the sentences to summarize this study and arrived at the final conclusion. At the same time, in the discussion section, we also added a focus on the basic results and conducted a dialogue with relevant literature to enrich the structure and content of the article and make it more complete. The relevant modifications have been revised in the revised version and highlighted in red . ]
Please see the attachment
Reviewer 2 Report
Comments and Suggestions for Authors
I think the study is well designed and presents interesting findings. However, I suggest the authors to revise the manuscript to make it stronger. Here are my suggestions:
1) I think the Introduction is little wordy. I think the authors can put some information presenting here to Literature Review. Then, they can spend space on highlighting the research gaps that they would like to address and how they actually address the gaps in the study. I think they should make more explicitly claim about the research gaps.
2) It is good to know that the study is informed by boundary theory. However, I would like to learn more about it and know how it inform the study. In the present version of the manuscript, boundary theory is just like a name here. But, the authors do not provide a critical review on it in order to help audience to know its contribution to the study and the contribution of the study to the theory.
3) I am focused by the research design. The authors say there were 125 participants, but there were 625 daily survey questionnaires. So, the sampling size is 125 or 625? Or it is a longitudinal study that had investigated 125 participants for 5 days so that it generated 625 questionnaires in total as longitudinal data? If it is a longitudinal study, why do not the authors analyze the longitudinal relationship between the variables? Why did they need to collect so many survey data with the same group of participants. How did they manage and analyze the data if they say it is not longitudinal research? Did the just accumulate all the responses from the 5 surveys and then compute the analysis? How do they justify such "interesting" use of the longitudinal data if it is longitudinal study?
4) The authors say "empirical sampling method". But, I never heard about it. I just know probability sampling and non-probability (or purposive) sampling. I am not sure whether the authors would like to say it is a probability sampling? If yes, please tell me which type of probability sampling was used? According to the manuscript, I cannot identify it is a probability sampling. It is just like a conventional or snowball sampling (a type of non-probability sampling), as the authors just personally contacted potential participants to join the study. The authors did not mean which cohort of MBA students they were investigated. This imply that they heavily reply on non-probability sampling approach for data collection. Otherwise, they should report, for example, "the study focused on MBA students from a Chinese university business school between cohorts 2020 and 2023. According to the business school's record, there were 600 students of the cohorts. Therefore, 235 students were sampled (95% confidence level, margin of error = 5%; and population proportion =50%). Based on the name list, the researcher team assigned a number for each student and then randomly selected the numbers. The selected students were contacted by email (or WeChat) in order to invite them to complete a questionnaire" Therefore, I encourage the authors provide more details about the sampling and data collection procedure.
Author Response
About the modified version,Please see the attachment
[Comments 1: I think the Introduction is little wordy. I think the authors can put some information presenting here to Literature Review. Then, they can spend space on highlighting the research gaps that they would like to address and how they actually address the gaps in the study. I think they should make more explicitly claim about the research gaps.]
Response 1: [Thank you for pointing this out. We agree with this comment. Based on your valuable suggestions, we have simplified the introduction section, highlighting the existing research gaps and the efforts we have made to address them. Specifically, we propose our actual work content by addressing the research gaps. This section has been revised in the introduction. Moreover, we have integrated the parts from the introduction into the literature review section.]
[Comments 2: It is good to know that the study is informed by boundary theory. However, I would like to learn more about it and know how it inform the study. In the present version of the manuscript, boundary theory is just like a name here. But, the authors do not provide a critical review on it in order to help audience to know its contribution to the study and the contribution of the study to the theory. ]
Response 2: [Thank you for pointing this out. We agree with this comment. Following your valuable suggestions, we have added important content to the literature review section - boundary theory and its specific applications in this study. This section can help you better understand boundary theory, and you can refer to it in the revised version.]
[Comments 3: I am focused by the research design. The authors say there were 125 participants, but there were 625 daily survey questionnaires. So, the sampling size is 125 or 625? Or it is a longitudinal study that had investigated 125 participants for 5 days so that it generated 625 questionnaires in total as longitudinal data? If it is a longitudinal study, why do not the authors analyze the longitudinal relationship between the variables? Why did they need to collect so many survey data with the same group of participants. How did they manage and analyze the data if they say it is not longitudinal research? Did the just accumulate all the responses from the 5 surveys and then compute the analysis? How do they justify such "interesting" use of the longitudinal data if it is longitudinal study? ]
Response 3: [We apologize for any confusion caused to you. This study is a longitudinal study that primarily involved a five-day survey of 125 participants. The reason for using longitudinal surveys is that most of the variables involved in this study have significant fluctuations at the daily level. For example, there are significant differences in the time and frequency of employees engaging in WCBA on a daily basis, which also leads to corresponding fluctuations in related mediator and outcome variables. In order to capture this difference well, this study fully draws on the latest research methods and research designs, namely the ESM research method, which is essentially a log survey study that can observe and questionnaire survey the same group of research subjects at the daily level for a continuous period of time to obtain relatively accurate and timely feedback from the subjects. The data obtained in this study further confirms that individuals have differences in their daily WCBA levels, indicating intra group variability. Therefore, this study draws on the approach of previous research and adopts the increasingly popular longitudinal survey method - ESM ]
[Comments 4: The authors say "empirical sampling method". But, I never heard about it. I just know probability sampling and non-probability (or purposive) sampling. I am not sure whether the authors would like to say it is a probability sampling? If yes, please tell me which type of probability sampling was used? According to the manuscript, I cannot identify it is a probability sampling. It is just like a conventional or snowball sampling (a type of non-probability sampling), as the authors just personally contacted potential participants to join the study. The authors did not mean which cohort of MBA students they were investigated. This imply that they heavily reply on non-probability sampling approach for data collection. Otherwise, they should report, for example, "the study focused on MBA students from a Chinese university business school between cohorts 2020 and 2023. According to the business school's record, there were 600 students of the cohorts. Therefore, 235 students were sampled (95% confidence level, margin of error = 5%; and population proportion =50%). Based on the name list, the researcher team assigned a number for each student and then randomly selected the numbers. The selected students were contacted by email (or WeChat) in order to invite them to complete a questionnaire" Therefore, I encourage the authors provide more details about the sampling and data collection procedure. ]
Response 4: [ We apologize again for the confusion caused to you, and it is our obligation to briefly introduce this method to you.Experience sampling method is an effective method of collecting longitudinal data. Experience Sampling Methodology (ESM) is a method of collecting instantaneous evaluations of events experienced by people in a short period of time multiple times and recording them. It uses repeated sampling to collect information that is easily influenced by time and individual factors. Its biggest feature is to collect individuals' immediate responses (including emotions, perceptions, attitudes, and evaluations) at multiple time points. In a word, experience sampling method refers to a set of data collection methods for gathering systematic self-reports of behaviors, emotions, or experiences as they occur in the individual’s natural environment. And In this study, it can actually be referred to as a Daily Diary survey.]
The attachment provides the modified version
Here are some relevant references of ESM.
1)Beal D. J. (2015). ESM 2.0: State of the art and future potential of experience sampling methods in organizational research. Annual Review of Organizational Psychology and Organizational Behavior, 2(1), 383–407.
2)Bolger, N., Stadler, G., & Laurenceau, J. P. (2012). Poweranalysis for intensive longitudinal studies. In M. R. Mehl& T. S. Conner (Eds.), Handbook of research methods for studying daily life (pp. 285–301). New York: Guilford Press.
3)Bolger, N., & Laurenceau, J. P. (2013). Intensive longitudinalmethods: An introduction to diary and experience sampling research . New York: Guilford Press.Mehl, M. R., & Conner, T. S. (2012). Handbook of researchmethods for studying daily life . New York: Guilford Press.
Reviewer 3 Report
Comments and Suggestions for Authors
The theoretical grounding for the inverted U shape association are still not fully developed. It is difficult to understand why one would expect an inverted U shape for PC but not for RC.
It should be clarified what PC and RC refer to. In the theoretical model and hypotheses PC and RC have different meanings than the ones specified in the abstract: "To clarify the complexity of the impact of WCBA on well-being at work, based on boundary theory, we divided WCBA into proactive WCBA
(PC) and reactive WCBA (RC), and examined the double-edged sword effect of WCBA on well- being at work, as well as the mediating mechanisms of job control and work-to-home conflict, and the moderated effects of boundary segmentation preferences."
In the theoretical section you mentioned: "Gong et al. (2024) argue that the key to the “Empowerment/Enslavement” Paradox lies in distinguishing between Personal Connectivity (PC) and Remote Connectivity (RC) [4]. The modes in which PC and RC alter boundaries differ, which determines the types of interruptions they cause and consequently their differing impacts on job control"
Finally in the hypotheses you refer to psychological contracts (PC) and role conflict (RC).
This is really unclear!
The second hypotheses confuses mediation with moderation. This is not acceptable in a scientific paper! See below
H2a: Work-to-home conflict mediates the positive relationship between PC (RC) and emotional exhaustion. The increase of PC (RC) will exacerbate work-to-home conflict, thereby diminishing employees' well-being at work.
H2b: Compared to PC, RC leads to a greater extent of work-to-home conflict, ultimately resulting in a more pronounced reduction in well-being at work.
Methods
The ICC values are unreasonably high ... I do not trust the results reported in your tables.
PC and RC are highly correlated, I think the inverted U shape is a statistical artifact - given the high correlation both variables should display similar relationships = this needs explanation
The sample is too small for a multilevel analysis - I do not think the analyses are appropriate
No regression results are presented - the quadratic effect cannot be inferred from your reported results!!!
Comments on the Quality of English Language
Please proof edit your manuscript
Author Response
Thank you for providing these valuable suggestions, which have greatly improved the quality of our paper.
The revised version of the manuscript is as follows, Please see the attachment.
Author's Notes
[Comments 1:[ The theoretical grounding for the inverted U shape association are still not fully developed. It is difficult to understand why one would expect an inverted U shape for PC but not for RC.]
Response 1: [Thank you for pointing this out. We have provided a detailed explanation regarding this issue. PC and RC have different boundary change patterns, which determine that they will cause different interruptions and have different impacts on job control. Interruption refers to the obstruction or delay of actors by breaking the continuity of ongoing activities, which can be divided into two types: external interruption (events in the environment causing changes in activities) and internal interruption (changes in activities are spontaneous). RC will definitely generate interruptions: whether it is switching from family activities to work or stopping work to respond to requests from other members in the work field, they are all external interruptions. The requirement for immediate response to work is unpredictable and mandatory, which can interfere with employees' time and space management. PC may not necessarily cause interruptions, such as planning a period of continuous work from home in advance, so that life and work processes will not be disturbed. Moreover, PCs only generate internal interruptions, which make employees feel that interruptions are of their own will and provide them with a sense of control in managing time and space.. RC means doing according to the requirements in the field of work. In summary, PC can enhance employees' job control, while RC will weaken employees' job control.
However, when PCs become excessive, employees may also begin to feel out of control over their work. With the extension of PC time, the sense of job control brought by actively initiating work is gradually offset. Employees are trapped in almost 24/7 work, becoming 'slaves' trapped in a' digital cage ', thus losing their agility and control. Employees are actually always online, which can weaken their autonomy in scheduling, work methods, and decision-making. At this point, adding more PCs will only weaken job control.
Therefore, PC has an inverted U-shaped impact on job control: when the level of PC is low, PC will enhance employees' job control; When the level of PC is high, PC will actually weaken employees' job control.]
Comments 2:[It should be clarified what PC and RC refer to. In the theoretical model and hypotheses PC and RC have different meanings than the ones specified in the abstract: "To clarify the complexity of the impact of WCBA on well-being at work, based on boundary theory, we divided WCBA into proactive WCBA(PC) and reactive WCBA (RC), and examined the double-edged sword effect of WCBA on well- being at work, as well as the mediating mechanisms of job control and work-to-home conflict, and the moderated effects of boundary segmentation preferences."In the theoretical section you mentioned: "Gong et al. (2024) argue that the key to the “Empowerment/Enslavement” Paradox lies in distinguishing between Personal Connectivity (PC) and Remote Connectivity (RC) [4]. The modes in which PC and RC alter boundaries differ, which determines the types of interruptions they cause and consequently their differing impacts on job control". Finally in the hypotheses you refer to psychological contracts (PC) and role conflict (RC).]
Response 2: [Thank you for pointing this out. Based on your suggestions, we have made detailed revisions and content additions in the introduction section. And all the meanings of PC and PC in the abstract, introduction, and model assumptions respectively refer to proactive WCBA(PC) and reactive WCBA (RC). This is a spelling and translation error on our part. All the meanings of PC and RC in the article respectively refer to proactive non working time work connected behavior (PC) and reactive non working time work connected behavior (RC). ]
[Comments 3:[The second hypotheses confuses mediation with moderation. This is not acceptable in a scientific paper! See below
H2a: Work-to-home conflict mediates the positive relationship between PC (RC) and emotional exhaustion. The increase of PC (RC) will exacerbate work-to-home conflict, thereby diminishing employees' well-being at work.
H2b: Compared to PC, RC leads to a greater extent of work-to-home conflict, ultimately resulting in a more pronounced reduction in well-being at work.]
Response 3: [Thank you for pointing this out. We have made detailed modifications to reduce misunderstandings
H2a: PC and RC is negatively related to well-being at work through increased work-to-home conflict, RC is negatively related to well-being at work through increased work-to-home conflict.
H2b: Compared to PC, RC leads to a greater extent of work-to-home conflict, ultimately resulting in a more pronounced reduction in well-being at work.]
[Comments 4:[The ICC values are unreasonably high ... I do not trust the results reported in your tables.]
Response 4: [Thank you for your question, but we must provide a response and explanation. This is related to the research method used in this study, which conducted a five-day survey of 125 employees. We obtained data from each employee's five-day survey, and for each individual (whose data report for five consecutive days is considered a group), it is well understood that the perception of the relevant variables for five consecutive working days within a week has a strong correlation. Therefore, the ICC (1) value of the data appears to be relatively high. ]
[Comments 5:[ PC and RC are highly correlated, I think the inverted U shape is a statistical artifact - given the high correlation both variables should display similar relationships = this needs explanation. ]
Response 5: [ We believe that there is a possibility that the biggest difference between PC and RC is whether the work connectivity behavior after work is initiated by employees or passively responded to work requirements after work,
In the workplace, PC and RC are actually strongly correlated, which is very common for Chinese employees and even difficult to avoid. Chinese employees often actively participate in work connectivity during off work hours, and are also required to keep their phones open during off work hours to respond to work demands at any time. Therefore, during a day, Chinese employees perceive both high PC and RC, but there are differences in the impact of PC and RC on their sense of job control.
In actual work situations, employee A proactively communicates with employee B at the end of the workday, and their response is highly likely to be PC. At the same time, employee A may also be required to respond to C's work requests, so employee A may also answer a higher degree of RC. When the number of connections or connection time is limited, the impact of employee A on job control will increase with the increase of enhanced PCs. However, excessive PCs. can also cause employee A to feel constrained. Therefore, when the PC exceeds a certain level, A's sense of job control will decrease with the increase of PC. However, at the same time, the increase in RC will cause a continuous decline in employee A's sense of job control. In summary, the sense of job control is influenced by both PC and RC, and the degree to which the same employee engages in both PC and RC may be high or low, depending on the culture of the company in the region. We have provided detailed responses in the theoretical assumptions section and Comment 1 regarding why RC and PC have different impacts on job control.
In order to further exclude the influence of RC on job control similar to PC, we also conducted supplementary validation studies to further verify the effect of RC squared on job control perception. The results showed that the coefficient of RC quadratic term was not significant. ]
[Comments 6:[The sample is too small for a multilevel analysis - I do not think the analyses are appropriate.]
Response 6: [For the empirical sampling method, this sample size can be guaranteed. Before aggregation, the actual sample size of the data was 125 individuals for five consecutive days, which reached 125 * 5=725. After aggregation, it reached 125 group of data. After referring to multiple authoritative journals, it was found that robustness results could still be obtained with fewer groups than ours. ]
[Comments 7:[No regression results are presented - the quadratic effect cannot be inferred from your reported results!!!.]
Response 7: [We forgot to label the estimated coefficients of the PC squared term (PC * PC) and RC squared term (RC * RC) in Figure 2. In the revised version, we have made supplements and added a relationship diagram between PC and job control, as shown in Figure 3 ]
Reviewer 4 Report
Comments and Suggestions for Authors
It seems that this a reviewd manuscript, since ther are pieces that have been flagged. I think it is enogh clear and readable to be published
Author Response
thanks for your work.
Reviewer 5 Report
Comments and Suggestions for Authors
Thank you for submitting this manuscript. The topic is relevant and interesting. I am including comments below for improvement of the paper.
1. The authors write: “First of all, this study examines the impact of WCBA on well-being at work, specifically dividing WCBA into PC) and RC at the behavioral aspect.”
Could you please clarify what PC and RC are earlier on (at first mention) in the Introduction, and please provide examples. I realize the authors provided details on these aspects in section 2.1; however, this information needs to be included earlier on in the paper – this provides the reader with greater clarity.
2. The authors write: “The study compares and analyzes the heterogeneous impacts of these two types of WCBA on well-being at work, aiming to provide explanations for the contradictions found in previous research.”
Could you please highlight some of the contradictions from previous research.
3. The authors write: “Secondly, this study reveals an inverted U-shaped effect of PC on job control and well-being 106 at work, offering a unique perspective on understanding the relationship between PC and well-being at work.”
This sounds like a result from your analysis. If this is the case, could you please move this to the results section. The introduction should just provide a background to the study without giving away any of the findings.
4. The objectives of the study need to be somewhat clearer to the reader. Therefore, could you please state the objectives of this study by using a numbered list.
5. The authors write: “Finally, this study examines the boundary conditions of work-to-home conflict. It contributes to a deeper understanding of individual differences in coping with RC.”
This is the first time that coping is mentioned in the paper. Could you please touch on coping earlier on in the introduction so that the text provides a segway/link with this sentence.
6. Section 2.2 – could you please remove the underlining and italicizing of the text.
7. Greater conciseness is needed for the introduction. I would suggest significantly condensing it and using the following structure for the introduction:
· * What is the issue you are looking to address and why is it important?
· * What does the literature say about this issue?
· * What are the objectives of your research?
8. A chunk of text in the Materials and Methods is in red, bolded font, and there are also highlights. Could you please check and undo this. Also, the conclusion to the paper is all in boded font – please undo and edit.
9. Table 1 heading – the following words do not need to be capitalized: “descriptive statistics”, “correlations”, “multilevel”.
10. The authors write: “this study selects MBA students from the finance, software services, and online education industries as participants, who are primarily from the business schools of five universities in China”
In the Discussion section, could you please discuss the limitations related to selection bias and generalizability. For example, this study is based only on MBA students, so it might not generalise to individuals who are not in a university setting. Also, the participants are quite young, so the findings might not generalise to older individuals.
11. In general, greater clarity is required throughout the paper.
Comments on the Quality of English LanguageMinor editing needed.
Author Response
[Comments 1:The authors write: “First of all, this study examines the impact of WCBA on well-being at work, specifically dividing WCBA into PC) and RC at the behavioral aspect.”
Could you please clarify what PC and RC are earlier on (at first mention) in the Introduction, and please provide examples. I realize the authors provided details on these aspects in section 2.1; however, this information needs to be included earlier on in the paper – this provides the reader with greater clarity.]
Response 1: [Thank you for pointing this out. Based on your suggestion, we have added this section to the introduction.]
Comments 2:[The authors write: “The study compares and analyzes the heterogeneous impacts of these two types of WCBA on well-being at work, aiming to provide explanations for the contradictions found in previous research.”Could you please highlight some of the contradictions from previous research. ]
Response 2: [Thank you for pointing this out. Based on your suggestion, we have added this section to the introduction. Based on the research of Gong et al. (2024) [4], this study categorizes WCBA into two types: proactive non-working time work connected behavior (PC, e.g. An employee came up with a good work idea while relaxing at home, so he gave up resting and went online to connect with work) and reactive non-working time work connected behavior (RC, e.g. Am employee received an instant message from the leader during dinner, so he interrupted family activities and went to work according to instructions. )]
[Comments 3:[The authors write: “Secondly, this study reveals an inverted U-shaped effect of PC on job control and well-being 106 at work, offering a unique perspective on understanding the relationship between PC and well-being at work.” This sounds like a result from your analysis. If this is the case, could you please move this to the results section. The introduction should just provide a background to the study without giving away any of the findings.]
Response 3: [Thank you for pointing this out. Based on your suggestion, we have revised this section in the introduction.]
[Comments 4:[ The objectives of the study need to be somewhat clearer to the reader. Therefore, could you please state the objectives of this study by using a numbered list. ]
Response 4: [Thank you for pointing this out. Based on your suggestion, we have revised this section in the introduction. ]
[Comments 5:[ The authors write: “Finally, this study examines the boundary conditions of work-to-home conflict. It contributes to a deeper understanding of individual differences in coping with RC.” This is the first time that coping is mentioned in the paper. Could you please touch on coping earlier on in the introduction so that the text provides a segway/link with this sentence.]
Response 5: [ We have reorganized this part of the content. ]
[Comments 6:[Section 2.2 – could you please remove the underlining and italicizing of the text.]
Response 6: [We corrected it in the revised manuscript. ]
[Comments 7:[ Greater conciseness is needed for the introduction. I would suggest significantly condensing it and using the following structure for the introduction:
- * What is the issue you are looking to address and why is it important?
- * What does the literature say about this issue?
- * What are the objectives of your research? ]
Response 7: [Thank you for pointing this out. Based on your suggestion, we have made significant revisions to the introduction and rephrased this section ]
[Comments 8:[A chunk of text in the Materials and Methods is in red, bolded font, and there are also highlights. Could you please check and undo this. Also, the conclusion to the paper is all in boded font – please undo and edit.]
Response 8: [We corrected it in the revised manuscript. ]
[Comments 9:[ Table 1 heading – the following words do not need to be capitalized: “descriptive statistics”, “correlations”, “multilevel”.]
Response 9: [We corrected it in the revised manuscript. ]
[Comments 10:[The authors write: “this study selects MBA students from the finance, software services, and online education industries as participants, who are primarily from the business schools of five universities in China”. In the Discussion section, could you please discuss the limitations related to selection bias and generalizability. For example, this study is based only on MBA students, so it might not generalise to individuals who are not in a university setting. Also, the participants are quite young, so the findings might not generalise to older individuals.]
Response 10: [Thank you for pointing this out. Based on your suggestion, we will add this content in the Limitations and Future Research section. And all MBA students have full-time jobs, just use weekends or holidays to complete their studies.]
Thank you for providing these valuable suggestions, which have greatly improved the quality of our paper.
The revised version of the manuscript is as follows, Please see the attachment.

Reviewer 6 Report
Comments and Suggestions for Authors
My comments are presented in the file attached to this message.

First and foremost, it would be more convenient for the readers if the subtitles included the full name of the constructs, not using abbreviations. Another alarming point in terms of abbreviations is that the authors use ‘PC’ for different constructs (proactive non-working time work connected behavior (see p.3), Personal Connectivity (see p.5) or psychological contract (PC) (see p.5 again line 229). The same applies for RC (Remote Connectivity (RC) (line 201, p.5; role conflict (RC) (line 233, p.5). Due to this confusion, Section 2.4 is not clear at all. The authors should provide consistency or connectivity between definitions with the same abbreviation.
Author Response
Author's Notes
Conments: First and foremost, it would be more convenient for the readers if the subtitles included the full name of the constructs, not using abbreviations. Another alarming point in terms of abbreviations is that the authors use ‘PC’ for different constructs (proactive non-working time work connected behavior (see p.3), Personal Connectivity (see p.5) or psychological contract (PC) (see p.5 again line 229). The same applies for RC (Remote Connectivity (RC) (line 201, p.5; role conflict (RC) (line 233, p.5). Due to this confusion, Section 2.4 is not clear at all. The authors should provide consistency or connectivity between definitions with the same abbreviation.I appreciate reviewing the manuscript titled ‘Differential Effects of Proactive and Reactive work connectivity behavior after-hours on Well-Being at Work: A Boundary Theory Perspective’. Overall, the idea of the study is interesting, especially exploring two types of work connectivity behaviors after-hours. However, it is difficult to follow this paper. First and foremost, it would be more convenient for the readers if the subtitles included the full name of the constructs, not using abbreviations. Another alarming point in terms of abbreviations is that the authors use ‘PC’ for different constructs (proactive non-working time work connected behavior (see p.3), Personal Connectivity (see p.5) or psychological contract (PC) (see p.5 again line 229). The same applies for RC (Remote Connectivity (RC) (line 201, p.5; role conflict (RC) (line 233, p.5). Due to this confusion, Section 2.4 is not clear at all. The authors should provide consistency or connectivity between definitions with the same abbreviation.
It would be better to use the same title for the investigated constructs (e.g., work control and
job control, see p.4)
Response: the same expression was performed for the investigated constructs. Thank you for pointing this out. Based on your suggestions, we have made detailed revisions and content additions in the introduction section. And all the meanings of PC and PC in the abstract, introduction, and model assumptions respectively refer to proactive WCBA(PC) and reactive WCBA (RC). This is a spelling and translation error on our part. All the meanings of PC and RC in the article respectively refer to proactive non working time work connected behavior (PC) and reactive non working time work connected behavior (RC). And work control is completely replaced by job control.
[Comments 1:What is meant by the “Empowerment/Enslavement” Paradox (see p.5)?]
Response 1: [Thank you for pointing this out. The "authorization-slavery paradox" means that WCBA, while enabling employees, makes them slaves who work all day and night and fail to get timely rest.]
Comments 2:The hypotheses are not always clearly articulated – think about positing positive or negative relationships between the variables of interest or splitting up several suggestions. That means the hypothesis meets the tests of good hypothesis writing; for example see Step-by-Step Guide: How to Craft a Strong Research Hypothesis
(https://scientific-publishing.webshop.elsevier.com/manuscript-preparation/whathow-write-good-hypothesis-research/). For example, H1 posits ‘Job control mediates the inverted U-shaped relationship between psychological contract (PC) and job satisfaction. Excessive or insufficient PC hinders employees from attaining job control, thereby reducing job satisfaction. Only a moderate level of PC maximizes employees' job control and subsequently enhances well-being at work.’]
Response 2: [Thank you for pointing this out. Based on your suggestion, We have made modifications to the expression of the research hypothesis. ]
[Comments 3:[3. Is stress conflict (line 242, p.5) and pressure conflict (line 240, p.5) synonyms? If so, mention this fact in the text. ]
Response 3: [Thank you for pointing this out. Based on your suggestion, we have standardized the expression of this phrase]
[Comments 4:[ What is meant by PCS NTWCB?]
Response 4: [it is PC, this is the mistake of our work.]
[Comments 5:[ 5.Please discuss more about well-being as a concept and a key variable in this study. ]
Response 5: [ The introduction and outcome discussion focus on discussing well being, and the main focus of this study is on the relationship between PC, RC, and well-being at work , so most of the content is focused on the description of the relationship. And well-being at work is described in detail in the second paragraph.]
[Comments 6:[Please describe how ‘MBA students from the finance, software services, and online education industries as participants, who are primarily from the business schools of five universities in China’ (line 337-339, p.7) can be a suitable sample for this study about work connectivity behaviors? You should describe at least their employment status (full-time/part-time) and work mode (remote, hybrid, or in office). ]
Response 6: [Thank you for pointing this out. They are in full-time work during normal working hours, working in a regular office. And, the study of MBA courses was conducted during holidays and weekends. ]
[Comments 7:[What is meant by NTWCB (line 355, p.8)?]
Response 7: [[it is WCBA, this is the mistake of our work. We corrected it in the revised manuscript. ]
[Comments 8:[ I cannot see this coefficient in Figure 2 “the square term of PC had a negative association with job control (β=-0.09, p<0.01)”.]
Response 8: [[We forgot to label the estimated coefficients of the PC squared term (PC * PC) and RC squared term (RC * RC) in Figure 2. In the revised version, we have made supplements and added a relationship diagram between PC and job control, as shown in Figure 3 ]
[Comments9:Please explain why you decided to use the square term of PC?]
Response 9: [Thank you for pointing this out. We have provided a detailed explanation regarding this issue. PC and RC have different boundary change patterns, which determine that they will cause different interruptions and have different impacts on job control. Interruption refers to the obstruction or delay of actors by breaking the continuity of ongoing activities, which can be divided into two types: external interruption (events in the environment causing changes in activities) and internal interruption (changes in activities are spontaneous). RC will definitely generate interruptions: whether it is switching from family activities to work or stopping work to respond to requests from other members in the work field, they are all external interruptions. The requirement for immediate response to work is unpredictable and mandatory, which can interfere with employees' time and space management. PC may not necessarily cause interruptions, such as planning a period of continuous work from home in advance, so that life and work processes will not be disturbed. Moreover, PCs only generate internal interruptions, which make employees feel that interruptions are of their own will and provide them with a sense of control in managing time and space.. RC means doing according to the requirements in the field of work. In summary, PC can enhance employees' job control, while RC will weaken employees' job control.
However, when PCs become excessive, employees may also begin to feel out of control over their work. With the extension of PC time, the sense of job control brought by actively initiating work is gradually offset. Employees are trapped in almost 24/7 work, becoming 'slaves' trapped in a' digital cage ', thus losing their agility and control. Employees are actually always online, which can weaken their autonomy in scheduling, work methods, and decision-making. At this point, adding more PCs will only weaken job control.
Therefore, PC has an inverted U-shaped impact on job control: when the level of PC is low, PC will enhance employees' job control; When the level of PC is high, PC will actually weaken employees' job control.]
Thank you for providing these valuable suggestions, which have greatly improved the quality of our paper.
The revised version of the manuscript is as follows, Please see the attachment.
Round 2
Reviewer 2 Report
Comments and Suggestions for Authors
Thanks for the positive response. Although you explain the methodology issue in details, you have made any changes in the manuscript to make the design clearly. I recommend you should revise the Method section in order to avoid those confusing messages.
Author Response
[Comments 1:Thanks for the positive response. Although you explain the methodology issue in details, you have made any changes in the manuscript to make the design clearly. I recommend you should revise the Method section in order to avoid those confusing messages.]
Response 1: [Thank you for pointing this out. We agree with this comment. Therefore, we revise the Method section in order to avoid those confusing messages. To avoid any unnecessary misunderstandings and to better present the research procedures, we have rewritten nearly all of the content in this section. And we mark our revisions in red]
Please see the attachment.

Reviewer 3 Report
Comments and Suggestions for Authors
I appreciate the work you put into answering my remarks, but unfortunately the quality and the clarity of the of the paper did not increase.
The quadratic effect is estimated for the intra-individual model, therefore it is based in 5 assessment points - this is not enough for estimating an inverted U shaped association. The sample size illustrated refers to the between individual effects, which are not estimated.
The results are not presented in a Table format so readers can understand what regressions were carried out and how the effects were estimated. The study should distinguish between the two types of effects generated through the design. Within as well as between individual effects have to be reported in the table.
Hypotheses remain unclear - see below the hypotheses that are not formulated accurately
H1a: Job control mediates the inverted U-shaped relationship between PC and well-being at work. Excessive or insufficient PC hinders employees from attaining job control, thereby reducing well-being at work. Only a moderate level of PC maximizes employees' job control and subsequently enhances well-being at work. = WHY DO WE NEED THE DETAILED DESCRIPTION OF THE QUADRATIC EFFECT HERE?
H1b: Job control mediates the negative impact of RC on well-being at work. An increase in weakens employees' job control, consequently diminishing well-being at work. = THE ADDITIONAL INFORMATION ON MEDIATION IS NOT NEEDED
H2a: PC and RC is negatively related to well-being at work through increased work-to-home conflict, RC is negatively related to well-being at work through increased work-to-home conflict. = THIS HYPOTHESIS HAS 2 SPECIFIC CLAIMS INCLUDED MEANING THAT IT REQUIRES 2 DIFFERENT EMPIRICAL TESTS
H2b: Compared to PC, RC leads to a greater extent of work-to-home conflict, ultimately resulting in a more pronounced reduction in well-being at work. = THIS EFFECT WAS NOT DIRECTLY TESTED AS DIFFERENCES IN EFFECT SIZES
H3a: The negative impact of PC (RC) on well-being at work through work-to-home conflict is mitigated by integrators. When integrators’ boundary integration preference is higher, the effect of PC (RC) on work-to-home conflict is weaker, and then the effect on well-being at work is weaker. = SECOND PART OF THE HYPOTHESIS IS UNCLEAR AND NOT NECESSARY
H3b: Relative to "RC→ work-to-home conflict → well-being at work" , the relation-ship "PC→ work-to-home conflict → well-being at work" is more mitigated by individual integration preferences. = UNCLEAR WHAT IS HYPOTHESIZED HERE AND MOST CERTAINLY THE DIFFERENCE IN EFFECT SIZE WAS NOT EMPIRICALLY ESTIMATED.
I am sorry to bring forward these negative points, but the impression is that the remarks I have made in the previous round were not taken seriously!
Comments on the Quality of English LanguageNo comments
Author Response
Comments 1: [ The quadratic effect is estimated for the intra-individual model, therefore it is based in 5 assessment points - this is not enough for estimating an inverted U shaped association. The sample size illustrated refers to the between individual effects, which are not estimated.The results are not presented in a Table format so readers can understand what regressions were carried out and how the effects were estimated. The study should distinguish between the two types of effects generated through the design. Within as well as between individual effects have to be reported in the table.]
Response 1: [Thank you for pointing this out. We conducted a multi-level path mechanism test on the entire model. Prior to the model test, the results of the ICC (1) test showed that it is suitable for constructing a multi-level model(Zhu H. T. Data Aggregation Adequacy Testing in Multilevel Research: A Critical Literature Review and Preliminary Solutions to Key Issues[J]. Advances in Psychological Science, 2020,28(8):1392⁃1408). Therefore, the coefficient and significance of the path mechanism can be used to determine(Gong S, Li H, Xia M, Zhu J. Are You Asked to Work Overtime? Exploring Proactive and Reactive Work Connectivity Behaviors After-hours and Their Multi-path Effects on Emotional Exhaustion[J]. Management Review, 2024, 36(2): 154-166.)And Our model validation process strictly follows the research of Gong et al. (2024). The comparison of the magnitude of the impact effects within and between groups is not the focus of our research. Our focus is on the size of the path coefficient, which is the path validation of the overall multi-level model. According to our research results, the estimated path coefficients of the multi-level path model can fully test the hypothesis. We believe that the research has already explained it clearly, and other anonymous reviewers have recognized our testing method. In short, our study does not require further regression analysis to present the results in the table. Because the overall multi-level path mechanism model can be used to test corresponding hypotheses.]
Comments 2: [Hypotheses remain unclear - see below the hypotheses that are not formulated accurately.]
Response 2: [Thank you for pointing this out.We agree with this comment. According to the suggestion, we have streamlined the expression of the research hypothesis content.
Thank you for providing these valuable suggestions, which have greatly improved the quality of our paper.
The revised version of the manuscript is as follows, Please see the attachment.
Reviewer 5 Report
Comments and Suggestions for Authors
Thank you for addressing my previous comments. I am including further feedback for the fine-tuning of this paper:
- Please correct the typo: “Am employee” should be “An employee” on page 1 of the introduction section.
- Please remove the numbers at the end of the introduction (1, 2, 3, 4).
- The authors write: “2)Secondly, this study reveals an inverted complex effect of PC on job control and well-being at work, offering a unique perspective on understanding the relationship between PC and well-being at work.”
Is this a finding from the study? If this is the case, then could you please move this to the results section of the manuscript.
- Please correct the typo – there should be a space after the word ‘domains’ (the authors write: “two domains(work and family)” ).
- Please clarify the second-last paragraph in the introduction section (3).
- Could you please remove the ‘empowerment/enslavement paradox’ text – these are lines 196-199 within the manuscript. The authors could instead point out that this is a contradiction (e.g. the fact that employees derive greater control from the use of technology at work, but at the same time, this control is eroded when employees may be expected to be constantly connected) – feel free to rephrase this.
- Before the literature review section, could you please provide a description as to why a literature review is needed (e.g. why would an evidence synthesis be needed in order to understand the study presented in the manuscript better?)
- Hypothesis 1A talks about the inverted U-shaped relationship. The authors need to mention earlier on why they believe the relationship would be U-shaped, and provide references if this is based on previous literature.
- It is unclear why quotation marks are used in H3B.
- In the Materials and Methods section (and other sections), please reference any segments where quotation marks are used (if text is derived from other sources).
- Section 4.1 The authors write: “r = −0.24, p < 0.001), Indicating”. This is a typo – the word ‘indicating’ in the segment above should have a small rather than capital ‘i’. Please have another glance over the entire manuscript and correct other typos.
- The literature review is quite lengthy and may benefit from greater conciseness. The authors are advised to have another glance over the paper to ensure good structure and clarity throughout.
Comments on the Quality of English Language
Some editing of the text is needed.
Author Response
[Comments 1:- Please correct the typo: “Am employee” should be “An employee” on page 1 of the introduction section.
- Please remove the numbers at the end of the introduction (1, 2, 3, 4).]
Response 1: [Thank you for pointing this out. Based on your suggestion, we have revised this]
Comments 2:[ - The authors write: “2)Secondly, this study reveals an inverted complex effect of PC on job control and well-being at work, offering a unique perspective on understanding the relationship between PC and well-being at work.”
Is this a finding from the study? If this is the case, then could you please move this to the results section of the manuscript. ]
Response 2: [Thank you for pointing this out.Thank you for your suggestion. This should be our mistake, We will replace 'reveals' with' test '.
Secondly, this study tests an inverted complex effect of PC on job control and well-being at work, offering a unique perspective on understanding the relationship between PC and well-being at work.]
[Comments 3:[- Please correct the typo – there should be a space after the word ‘domains’ (the authors write: “two domains(work and family)” ).
- Please clarify the second-last paragraph in the introduction section (3).]
Response 3: [Thank you for pointing this out. Based on your suggestion, we have revised this ]
[Comments 4:[ - Could you please remove the ‘empowerment/enslavement paradox’ text – these are lines 196-199 within the manuscript. The authors could instead point out that this is a contradiction (e.g. the fact that employees derive greater control from the use of technology at work, but at the same time, this control is eroded when employees may be expected to be constantly connected) – feel free to rephrase this. ]
Response 4: [Thank you for pointing this out. Based on your suggestion, we have revised this ]
[Comments 5:[ - Before the literature review section, could you please provide a description as to why a literature review is needed (e.g. why would an evidence synthesis be needed in order to understand the study presented in the manuscript better?) ]
Response 5: [ Thank you for pointing this out. Based on your suggestion, we have revised this.
There have been several related achievements in the subdivision of WCBA and related explanatory theories, such as the classification of WCBA into PC and RC, which can be interpreted from the boundary theory to understand the influence of WCBA[4]. Reviewing these studies can help us understand the reasoning behind the hypotheses of this study. ]
[Comments 6:[- Hypothesis 1A talks about the inverted U-shaped relationship. The authors need to mention earlier on why they believe the relationship would be U-shaped, and provide references if this is based on previous literature.
- It is unclear why quotation marks are used in H3B.
- In the Materials and Methods section (and other sections), please reference any segments where quotation marks are used (if text is derived from other sources).]
Response 6: [We corrected it in the revised manuscript.
line218-219, Therefore, excessive PC will weaken job control. Thus, PC has an inverted U-shaped effect on job control[4]. ---We have added a reference.
-----And we removed the quotation marks]
[Comments 7:[- Section 4.1 The authors write: “r = −0.24, p < 0.001), Indicating”. This is a typo – the word ‘indicating’ in the segment above should have a small rather than capital ‘i’. Please have another glance over the entire manuscript and correct other typos. ]
Response 7: [Thank you for pointing this out. Based on your suggestion, we have made significant revisions ]
[Comments 8:[- The literature review is quite lengthy and may benefit from greater conciseness. The authors are advised to have another glance over the paper to ensure good structure and clarity throughout.]
Response 8: [Thank you for your suggestion. The initial version of the review is more concise, but some anonymous reviewers believe that it needs more rich content, so we have added and refined some content. We will carefully consider your significance. Thank you very much for your targeted feedback. ]
Thank you for providing these valuable suggestions, which have greatly improved the quality of our paper.
The revised version of the manuscript is as follows, Please see the attachment.

Reviewer 6 Report
Comments and Suggestions for Authors
Dear authors,
Thank you for the opportunity to review your manuscript titled ‘Differential Effects of Proactive and Reactive work connectivity behavior after-hours on Well-Being at Work: A Boundary Theory Perspective’. I have the following comments regarding the updated version:
-
Your hypotheses include several statements, which is confusing because it is unclear what you expect to learn about a phenomenon. For example: ‘H1a: Job control mediates the inverted U-shaped relationship between PC and well-being at work. Excessive or insufficient PC hinders employees from attaining job control, thereby reducing well-being at work. Only a moderate level of PC maximizes employees' job control and subsequently enhances well-being at work.’ (see p. 5). Present each of your hypotheses as a brief statement and avoid wordiness.
This comment applies to all your hypotheses, namely H1a, H1b, H2a, H2b, H3a, H3b.
Previously, I sent you the link to Step-by-Step Guide: How to Craft a Strong Research Hypothesis.
-
Hypotheses should be visualized in Figure 1—theoretical model.
-
There are some typos in the text ( see for example, ‘e.g. Am employee received an instant message from the leader during’ line 92 p.2). Proofreading is needed.
I wish you every success with this manuscript.
Comments on the Quality of English LanguageThere are some typos in the text ( see for example, ‘e.g. Am employee received an instant message from the leader during’ line 92 p.2). Proofreading is needed.
Author Response
[Comments 1:Your hypotheses include several statements, which is confusing because it is unclear what you expect to learn about a phenomenon. For example: ‘H1a: Job control mediates the inverted U-shaped relationship between PC and well-being at work. Excessive or insufficient PC hinders employees from attaining job control, thereby reducing well-being at work. Only a moderate level of PC maximizes employees' job control and subsequently enhances well-being at work.’ (see p. 5). Present each of your hypotheses as a brief statement and avoid wordiness.This comment applies to all your hypotheses, namely H1a, H1b, H2a, H2b, H3a, H3b. Previously, I sent you the link to Step-by-Step Guide: How to Craft a Strong Research Hypothesis. ]
Response 1: [Thank you for pointing this out.Thank you for your suggestion. We have made the necessary modifications as requested.]
Comments 2:Hypotheses should be visualized in Figure 1—theoretical model.]
Response 2: [Thank you for pointing this out. Based on your suggestion, We have made modifications to this. ]
[Comments 3:[There are some typos in the text ( see for example, ‘e.g. Am employee received an instant message from the leader during’ line 92 p.2). Proofreading is needed. ]
Response 3: [Thank you for pointing this out. Based on your suggestion, we have standardized the expression of this phrase]
Thank you for providing these valuable suggestions, which have greatly improved the quality of our paper.
The revised version of the manuscript is as follows, Please see the attachment.
